# FEDERATED BINARY MATRIX FACTORIZATION USING PROXIMAL OPTIMIZATION

## ABSTRACT

Identifying informative components in binary data is an essential task in many research areas, including life sciences, social sciences, natural language processing, and recommendation systems. Boolean matrix factorization (BMF) is a family of methods that performs this task by efficiently factorizing the data into its constituent parts. In real-world settings, the data is often distributed across stakeholders and required to stay private, prohibiting the straightforward application of BMF. To adapt BMF to this context, we approach the problem from a federated-learning perspective, while building on a state-of-the-art continuous binary matrix factorization relaxation to BMF that enables efficient gradient-based optimization. We propose to only share the relaxed component matrices, which are aggregated centrally using a proximal operator that regularizes for binary outcomes. We show the convergence of our federated proximal gradient descent algorithm and provide differential privacy guarantees. Our extensive empirical evaluation demonstrates that our algorithm outperforms, in terms of quality and efficacy, federation schemes of state-of-the-art BMF methods on a diverse set of real-world and synthetic data.

## 1 INTRODUCTION

Discovering patterns and dependencies in distributed binary data sources is a common problem in many applications, such as cancer genomics (Liang et al., 2020), recommendation systems (Ignatov et al., 2014), and neuroscience (Haddad et al., 2018). In particular, this binary data is often distributed horizontally (i.e., the rows of the data matrix are split across hosts) and may not be pooled. For example in cancer genomics, a common set of gene expressions is measured for biopsies performed in various hospitals, where data privacy mandates that the measurements may not be shared.

Standard variants of matrix factorization, such as Singular Value Decomposition (Golub and Loan, 1996), Principal Component Analysis (Golub and Loan, 1996), and Non-negative Matrix Factorization (NMF) (Paatero and Tapper, 1994; Lee and Seung, 1999; 2000) find patterns in real-valued data, but do not achieve interpretable results for binary data. Therefore, recent methods on Federated Non-negative Matrix Factorization (Li et al., 2021) that do not share data cannot be applied to most binary problems under privacy considerations.

For *centralized data*, Boolean matrix factorization (BMF) seeks to approximate a Boolean target matrix $A \in \{0, 1\}^{n \times m}$ by the Boolean product

$$A \approx [U \circ V]_{ij} = \bigvee_{l \in [k]} U_{il} V_{lk}$$

of two low-rank Boolean factor matrices (Miettinen et al., 2008), $U \in \{0, 1\}^{n \times k}$ (*feature matrix*) and $V \in \{0, 1\}^{k \times m}$ (*coefficient matrix*). Although there are myriad heuristics to approximate this NP-hard problem, doing so for *distributed data* without sharing private information remains an open problem. Directly applying federated-learning paradigms to BMF will factorize locally and aggregate centrally, requiring suitable aggregators that yield valid aggregations, including, for example, *rounded average*, *majority vote*, or *logical* `or`. We depict the impact of naïve yet valid aggregations in Fig. 1(a), which highlights that even the best combination of a local factorization algorithm and an aggregation scheme —here, ASSO (Miettinen et al., 2008) using logical `or`—leads to bad reconstructions on a toy example.

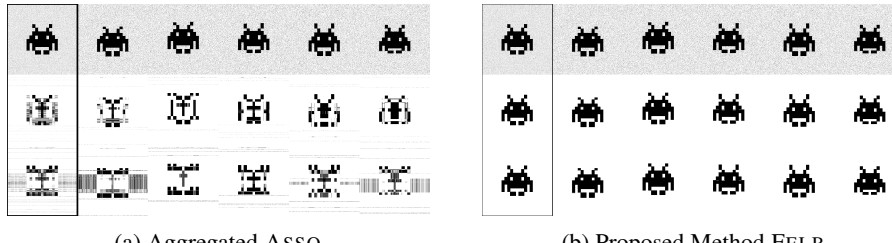

(a) Aggregated ASSO                                    (b) Proposed Method FELB

Figure 1: Our method reconstructs the input well. Representing 1s as black pixels, for ASSO using logical `or` (a) and our novel federated factorization called FELB (b), we show client-data—subjected to additive noise—(top), localized reconstructions (middle), and aggregation-based reconstructions (bottom). The box contains centralized combinations of the respective row.

Recently, it was shown that continuously relaxing BMF into a regularized *binary* matrix factorization problem using linear algebra (instead of Boolean) and proximal gradients, efficiently yields state-of-the-art performance (Dalleiger and Vreeken, 2022). Taking advantage of this approach, we propose a novel federated proximal-gradient method, FELB, that centrally, yet privacy-consciously aggregates non-sensitive coefficients using a proximal-averaging aggregation scheme. As illustrated in Figure 1(b), FELB achieves a nearly perfect reconstruction on the toy example. We demonstrate that our approach converges for a strictly monotonically increasing regularization rate. Moreover, we show that applying the Gaussian mechanism (Balle and Wang, 2018) guarantees differential privacy, and we empirically validate that the utility remains high. We show that FELB outperforms baselines derived via straightforward parallelizations of state-of-the-art BMF methods on a large number of datasets. Furthermore, parallelization *via* FELB allows us to scale up BMF even if data is centralized: We are able to factorize very large matrices efficiently and accurately, outperforming the state-of-the-art.

In summary, our main contributions are as follows:

- We present FELB, a novel federated proximal-gradient-descent for BMF.
- We introduce an adaptive variant, FALB, improving regularization over the state-of-the-art.
- We prove convergence and differential privacy guarantees for FELB.
- We experimentally show that FELB and FALB federatedly factorize distributed Boolean matrices efficiently and accurately.

## 2  RELATED WORK

To the best of our knowledge, there are no federated BMF algorithms. Our approach, however, is closely related to BMF and federated factorization.

We distinguish two classes of BMF methods: First, *discrete methods* that use Boolean algebra, such as ASSO (Miettinen et al., 2008) using a set-cover-like approach, GRECOND (Belohlávek and Vychodil, 2010), MEBF (Wan et al., 2020) using fast geometric segmentations, or SOFA (Neumann and Miettinen, 2020) based on streaming clustering. Second, *continuous methods* that use linear algebra for solving the binary matrix factorization problem, introduced by Zhang et al. (2007), and advanced further by Araujo et al. (2016) based on thresholding, and by Hess et. al (Hess et al., 2017; Hess and Morik, 2017) using a proximal operator. Combining ideas from the two complementary regularization strategies of Hess et al. (2017) and Zhang et al. (2007), Dalleiger and Vreeken (2022) recently proposed a proximal operator for its elastic-net-based regularizer.

As regards federated factorization, 'parallel' algorithms for matrix factorization (Yu et al., 2014) as well as binary matrix factorization (Khanna et al., 2013) seek computational efficiency while disregarding privacy aspects. The problem of matrix factorization for privacy-sensitive distributed data has been addressed by the federated-learning community with approaches for federated matrix factorization (Du et al., 2021) and federated non-negative matrix factorization (Li et al., 2021). These methods are, however, not specialized to Boolean matrices. In this work, we seek to close the remaining research gap, addressing the need for a federated, privacy-preserving binary (or Boolean)

matrix factorization algorithm. Recently, a federated proximal optimization for expensive to compute proximal operators Mishchenko et al. (2022) has been proposed, which is not needed for our operator.

# 3 PROXIMAL BINARY MATRIX FACTORIZATION

Having contextualized the high-level problem, we now introduce our binary matrix factorization, starting with the preliminary Boolean matrix factorization. Rather than relying on linear algebra, BMF uses Boolean algebra. Most importantly, this means that the product of two Boolean matrices $U \in \{0, 1\}^{n \times k}$ and $V \in \{0, 1\}^{k \times m}$ follows the algebra of a Boolean semi-ring $(\{0, 1\}, \vee, \wedge)$ and is defined as

$$[U \circ V]_{ij} = \bigvee_{l \in [k]} U_{il} V_{lj} \ , \tag{1}$$

where $U \circ V \in \{0, 1\}^{n \times m}$. This is identical to the standard outer product on a field where addition obeys $1 + 1 = 1$. For a given matrix $A \in \{0, 1\}^{n \times m}$ and a rank $k \in \mathbb{N}_{\leq \min\{n, m\}}$, BMF seeks to minimize

$$\|A - U \circ V\|_F^2 = \sum_{ij} A_{ij} \vee [U \circ V]_{ij} \tag{2}$$

in terms of Boolean factor matrices $U \in \{0, 1\}^{n \times k}$ and $V \in \{0, 1\}^{k \times m}$. As this problem is NP-complete (Miettinen and Neumann, 2020), solving it exactly is challenging, even for relatively small matrices. To solve this problem in practice for large matrices, we resort to a continuous relaxation into a *binary matrix factorization* problem that minimizes

$$\|A - UV\|_F^2 + R(U) + R(V) \ , \tag{3}$$

with a binary-inducing regularizer $R : \mathbb{R}^{n' \times m'} \to \mathbb{R}$, enabling efficient gradient-based optimizations.

Here, we first introduce a local algorithm for a single client (addressing federated BMF for multiple clients in Section 4). A regularizer that encourages binary solutions combines two elastic-nets (centered at 0 and 1, resp.),

$$R_{\kappa\lambda}(X) = \sum_{x \in X} \min \left\{ \kappa \|x\|_1 + \lambda \|x\|_2^2, \kappa \|x - 1\|_1 + \lambda \|x - 1\|_2^2 \right\} \tag{4}$$

into the almost W-shaped ELB regularizer (Dalleiger and Vreeken, 2022), yielding the gradient-based alternating update rules

$$U_t = \arg\min_U \|A - UV_{t-1}\|_F^2 + R(U) \text{ and } V_t = \arg\min_V \|A - U_{t-1}V\|_F^2 + R(V) \ ,$$

minimized by proximal-gradient updates $U_{t+1} = \text{prox}_{\kappa\lambda_t}(U_t - \eta_t \nabla_U \|A - U_t V_t\|_F^2)$ and $V_{t+1} = \text{prox}_{\kappa\lambda_t}(V_t - \eta_t \nabla_V \|A - U_t V_t\|_F^2)$, respectively, for the given proximal operator

$$\text{prox}_{\kappa\lambda} = (1 + \lambda)^{-1} \begin{cases} x - \kappa \, \text{sign}(x) & \text{if } x \leq \frac{1}{2} \\ x - \kappa \, \text{sign}(x - 1) + \lambda & \text{otherwise} \end{cases} \ , \tag{5}$$

where the step-size function $\eta_t$ is given. To improve convergence and differential privacy, we wrap this operator into a unit-cube proximal operator—similar to PRIMP (Hess et al., 2017)—thus clamping values in between 0 and 1.

## 3.1 ADAPTIVE REGULARIZATION RATES

Our proximal operator alone does not yield valid factorizations, but rather relies on a monotonically increasing user-defined regularization-rate $\lambda_t$ that increasingly-strongly nudges factors towards a valid BMF solution (Dalleiger and Vreeken, 2022). Ideally, in practice, we would like to select an individual $\lambda_t$ per client-data, which imposes a tuning burden on each client. To alleviate the complexity, we propose a novel variant of the ELB-regularizer that *adaptively* sets the regularization rate $\lambda_t(x)$ for each cell in each factor matrix, obtaining the ALB-regularizer, detailed in Appendix A. The closer a matrix cell $x$ gets to 0 (resp. 1), the higher the adaptive rate $\lambda_t(x)$ gets, thus pushing harder and improving convergence toward Boolean matrices. We achieve this by replacing $\lambda_t$ with $\lambda_t(x)$, giving rise to an adaptive proximal operator analogous to Eq. (5), where the adaptive rate $\lambda_t(x) = \lambda_t \cdot \sigma(x)$ is a sigmoidally-scaled regularization rate $\lambda_t$.

---

**Algorithm 1:** Federated Binary Matrix Factorization with FELB

---

**Input:** distributed target matrices $A^1, \ldots, A^C$, component-count $k$
**Output:** local feature matrices $U^1, \ldots, U^C$, global coefficient matrix $V$

1   initialize $U^i, V^i$ for $i \in [C]$ uniformly at random
2   **Locally** *at client $i$ in iteration $t$* **do**
3      abbreviate $a, u, v \leftarrow A^i, U^i, V^i$
4      $u \leftarrow \mathrm{prox}_{\kappa\lambda_t}(u - \eta_t^u \nabla_u \|a - uv\|_F^2)$
5      $v \leftarrow \mathrm{prox}_{\kappa\lambda_t}(v - \eta_t^v \nabla_v \|a - uv\|_F^2)$
6      **if** $t \bmod b = b - 1$ **then**
7          transmit $v$ to the server
8          receive $\widehat{V}$ from the server
9          let $v \leftarrow \widehat{V}$
10   **At server** *in round $t$* **do**
11      receive $V^1, \ldots, V^C$
12      aggregate $\widehat{V} \leftarrow \mathrm{prox}_{\kappa\lambda}\left(\frac{1}{C}\sum_{i=1}^C V^i\right)$
13      transmit $\widehat{V}$ to each client

---

# 4   FEDERATED PROXIMAL BINARY MATRIX FACTORIZATION

Next, we address federated BMF for multiple clients. For a given distributed matrix $A \in \{0,1\}^{n\times m}$,

$$\exists A^1, \ldots, A^C \in \{0,1\}^{\frac{n}{C}\times m} : A = \left[A^1, \cdots, A^C\right]^\top \ ,$$

over $C \in \mathbb{N}$ clients, we seek to find a distributed factorization using *local* matrices $U^i \in \{0,1\}^{\frac{n}{C}\times k}$ and a single *shared* matrix $\widehat{V} \in \{0,1\}^{k\times m}$, i.e., $A^i \approx U^i \circ \widehat{V}$. This partial sharing improves the privacy of subjects by withholding potentially private subject-feature relationships ($U^i$), and it also enables the discovery of an informative shared coefficient matrix ($\widehat{V}$). Without the knowledge of $U^i$, the server *cannot* estimate specific attributes of individual users (assuming sufficiently large client datasets). However, the server *can* estimate sets of commonly co-occurring attributes across all clients, such as common combinations of genetic markers that are indicative of a disease.

To obtain this desirable shared matrix $\widehat{V}$, we need to combine local $V^i$ matrices. However, doing so using common approaches (that are applicable to relaxed Boolean matrices) does not necessarily yield valid results: naïve averaging results in aggregants that are far from being binary, thus hindering convergence. Addressing this aggregation problem, we propose to push the average of $V^1, \ldots, V^C$ toward a valid outcome in terms of the regularizer from Eq. (4) (or its adaptive counterpart, Appendix A, Eq. (9)). We call the combination of this proximal aggregation with local proximal-gradient optimization steps via Eq. (3) the FELB algorithm (and its adaptive variant the FALB algorithm), detailed in Alg. 1: Local factors $U^i, V^i$ are initialized uniformly at random (line 1), and at each client in round $t$ (line 2), the local factor matrices are updated using the approach detailed in Section 3 (lines 4 and 5). Every $b$ rounds, the local matrices $V^i$ are send to the server (line 7). At the server, all local $V^i$ are received (line 11), aggregated by averaging the matrices and applying the proximal-operator (line 12). The aggregate is then send to all clients (line 13). Upon receiving the aggregate (lines 8 and 9), the client continues with another iteration. The proximal operator encourages convergence to a binary solution, as we show next.

## 4.1   CONVERGENCE ANALYSIS

Here, we show that FELB converges to a stable *binary* solution that does not change from one iteration to the next, i.e., $U_t = U_{t-1}$ and $V_t = V_{t-1}$, for $t \to \infty$, resulting in a valid BMF upon convergence.

**Proposition 1.** *For a matrix $A$ and a loss function $\ell$, assume that its gradients w.r.t. matrices $U$ and $V$ are bounded, i.e., there exists $R \in \mathbb{R}$ such that for all $U, V$ it holds that $\|\nabla_U \ell(A, U, V)\|, \|\nabla_V \ell(A, U, V)\| \le R$. If $\lambda_t : \mathbb{N} \to \mathbb{R}_+$ is strictly monotonically increasing, then FELB converges for a step-size $\eta_t \propto \lambda_t^{-1}$.*

*Proof.* In each update round, the client performs the proximal alternating linear minimization steps laid out in Eq. 3, yielding an updated $V_t$ (resp. $U_t$). Focusing on $V$ (independent of client-server communication), we first show that the gradient of $V$ goes to zero. Assuming an $R$-bounded gradient (i.e., $\|\nabla_V \nabla_V \|A - U_{t-1}V\|_F^2\| \leq R \in \mathbb{R}$), we know that

$$\lim_{t\to\infty} \left\|\eta_t \nabla_V \|A - U_{t-1}V\|_F^2\right\| = \lim_{t\to\infty} \eta_t \left\|\nabla_V \|A - U_{t-1}V\|_F^2\right\| \leq \lim_{t\to\infty} \eta_t R = 0 \ ,$$

since $\lambda_t$ is strictly monotonically increasing.

Second, we need to verify that the proximal operator projects to binary solutions, i.e., $\lim_{\lambda_t \to \infty} \text{prox}(x) \in \{0, 1\}$ for $\lambda_t \to \infty$. We do this with a case distinction: For $x \leq 0.5$, we obtain $\lim(x - \kappa \operatorname{sign}(x))(1 + \lambda_t)^{-1} = 0$ , and analogously for $x > 0.5$, we obtain $\lim(x - \kappa \operatorname{sign}(x - 1) + \lambda_t)(1 + \lambda_t)^{-1} = 1$ , thus having ensured a binary proximity, for $\lambda_t \to \infty$, any bounded $x$, and finite $\kappa \in \mathbb{R}_+$. Therefore, independent of communication rounds, the gradient converges to 0 and the proximal operator converges to a binary solution. It remains to show that for $t \to \infty$, a binary solution stays stable, meaning that a global binary solution implies local convergence. By assuming that a client in round $t$ receives a binary aggregate $\widehat{V}$ from the server, we obtain $\|\eta_t \nabla_V \|A - U_{t-1}V\|_{max}\| \leq \epsilon$ for $\epsilon < {}^1/2$, for an appropriate step-size $\eta_t \propto \lambda_t^{-1}$. By abbreviating the gradient-step result $V' = V_{t-1} - \eta_t \nabla_V \|A - U_{t-1}V\|_F^2$ we see that $V'_{ij} < {}^1/2$ if $V_{t-1} = 0$, and $V'_{ij} > {}^1/2$ if $V_{t-1} = 1$, which implies that $\text{prox}_{\lambda_t \kappa}(V')$ is binary and $V_t = V_{t-1}$. □

## 4.2 Differentially Private Felb

The proposed approach only shares coefficient matrices, so that no direct relationships between observations are shared. An attacker or a curious server can, however, attempt to infer local data from coefficients $V^i$. To guarantee differential privacy, we consider additive noise mechanisms. In a nutshell, in each iteration, each client adds noise before it transmits $V^i$ to the server. We consider the commonly-used Gaussian or Laplacian mechanisms, which—as the name implies—differ in the noise distribution. More formally, the Gaussian mechanism (Noble et al., 2022; Geyer et al., 2017) achieves $(\epsilon, \delta)$-differential privacy:

**Definition 1** (Dwork et al. (2014)). *For $\epsilon, \delta > 0$, a randomized algorithm $\mathcal{A} : \mathcal{X} \to \mathcal{Y}$ is $(\epsilon, \delta)$-differentially private if for all pairs of neighboring inputs $X, X'$ and for every subset $S \subset \mathcal{Y}$ it holds that*

$$P\left(\mathcal{A}(X) \in S\right) \leq e^\epsilon P\left(\mathcal{A}(X') \in S\right) + \delta \ .$$

Applying Gaussian noise with 0 mean and $\sigma$ variance to the local coefficients $V^i$ before sending ensures $(\epsilon, \delta)$-differential privacy for $\sigma = \Delta \epsilon^{-1} \sqrt{2 \log(5/(4\delta))}$ (Balle and Wang, 2018), where

$$\Delta = \sup_{X, X'} \|\mathcal{A}(X) - \mathcal{A}(X')\|$$

is the sensitivity of $\mathcal{A}$. To ensure bounded sensitivity, we clip all $V^i$ with clipping threshold $\theta > 1$ (Noble et al., 2022). Similarly, adding 0-mean $\Delta \epsilon^{-1}$-variance Laplacian noise achieves $(\epsilon, 0)$-differential privacy (Dwork et al., 2006).

## 5 Experiments

**Competitors.** Given that there are no *federated* matrix factorization algorithms that are tailored to binary data, we compare our approaches to *local* methods, whose outcomes are then partially transmitted to a central location and collectively aggregated, following established ad-hoc federation strategies. In particular, we adapt the localized algorithms, covering the state of the art in the method families (1) *cover-based Boolean matrix factorizations* (Asso, Miettinen et al. (2008); Grecond, Belohlávek and Vychodil (2010); mebf, Wan et al. (2020)) and (2) *relaxation-based binary matrix factorizations* (Zhang, Zhang et al. (2007); and Elbmf, Dalleiger and Vreeken (2022)), to factorize *distributed* matrices—factorizing locally and aggregating the coefficient matrices centrally, replacing the local coefficients. To ensure binary results, we employ three *aggregation strategies* designed to maintain valid matrices:

$$\begin{array}{ccc} \text{Rounded Average (6)} & \text{Majority Vote (7)} & \text{Logical }\mathtt{Or}\text{ (8)} \\[4pt] \left\lfloor C^{-1} \sum_{c \in [C]} V^c \right\rceil & \left[\sum_{c \in [C]} V^c_{ij} \geq C/2\right]_{ij} & \bigvee V^1, \ldots, V^C \ . \end{array}$$

We summarize the federation scheme for competing BMF methods used in our experiments, which leverages these aggregation methods, as Alg. 2 in Appendix B.

**Experiments.** Having identified our set of methods to compare to, we now describe our diverse set of experimental setup. First, we ascertain that FELB works reliably on synthetic data. Second, we empirically assess the differential-privacy properties of FELB. And third, we verify that FELB performs well on diverse real-world datasets drawn from four different scientific areas. To quantify the results, we report the relative reconstruction *loss* as the mean distance $C^{-1} \sum \frac{\|A^c - U^c \circ \widehat{V}\|}{\|A^c\|}$ , the *recall* as the mean number of identical 1s $C^{-1} \sum_c \|[A_{ij}^c = U_{ij}^c \circ \widehat{V}_{ij}]_{ij}$ , the *similarity* as the average number of identical bits $C^{-1} \sum_c \sum [A^c \odot (U^c \circ \widehat{V})]$ , and the *elapsed runtime* in seconds of each algorithm.

We implement FELB in the Julia language and run experiments on 32 CPU Cores of an AMD EPYC 7702 or one NVIDIA A40 GPU, reporting wall-clock time in seconds. We provide the source code, datasets, synthetic dataset generator, and other information needed for reproducibility.[1] In all experiments, we limit each algorithm run to 12h in total. We quantify the performance of *federated* ASSO, GRECOND, ELBMF, MEBF, FELB, and FALB in terms of loss, recall, similarity, and runtime, reporting results for *majority voting* in the following, and for *rounded averaging* and *logical* or in the Appendices D.1–D.5.

## 5.1 EXPERIMENTS ON SYNTHETIC DATA

In our experiments on synthetic data, we aim to answer the following questions:

**Q1** How robust are the algorithms in the context of noise?
**Q2** How scalable are the algorithms with increasing client counts?
**Q3** How achievable is differential privacy?

To answer these questions, we need a controlled test environment, which we construct by sampling random binomial-noise matrices into which we insert random densely populated ($\approx 90\%$) 'tiles'.

### 5.1.1 ROBUSTNESS REGARDING NOISE

To study the impact of noise on the quality of reconstructions, we generated synthetic matrices, introducing varying degrees of destructive XOR noise, ranging from $0\%$ (no noise, consisting solely of high-density tiles) to a maximum of $50\%$ (completely random noise). Employing a fixed number of 10 clients, we applied federated ASSO, GRECOND, MEBF, ELBMF, and ZHANG, alongside FELB and FALB to each dataset. We present loss, recall, similarity, and runtime in Fig. 2 and Appendix D.1.

Fig. 2(a) shows that FELB and FALB are the only algorithms that achieve robust reconstructions even at low noise levels. The loss steadily and smoothly increases to converge to 1 as the noise increases. This shows the ability of FELB and FALB to discern pure noise from meaningful signal. Additional evidence is the smooth decline in similarity (Fig. 2(b)) and recall (Fig. 2(c)), where FELB and FALB also consistently outperform ASSO, GRECOND, MEBF, ZHANG, and ELBMF. As illustrated in Fig. 2(d), GRECOND's runtime suffers severely from increasing noise levels. Conversely, the remaining methods exhibit only minor fluctuations in runtime, without showing any discernible trend.

### 5.1.2 SCALABILITY REGARDING CLIENTS

Next, we analyze the scalability of federated ASSO, GRECOND, ELBMF, MEBF, and ZHANG under *majority voting*, as well as of FELB and FALB, for varying numbers of clients, considering two contrasting scenarios of *scarce* and *abundant* data.

**Data Scarcity.** First, we study the case in which data is scarce, meaning that *data will not increase* with increasing number of clients. For this, we generate synthetic data with a fixed size, which we then distribute to an increasing number of clients, uniformly, meaning that an increase in clients will result in *smaller local datasets*, whose resulting high-dimensional few-observations regime poses a significant challenge for each client.

---

[1]Appendix C; DOI: Anonymized

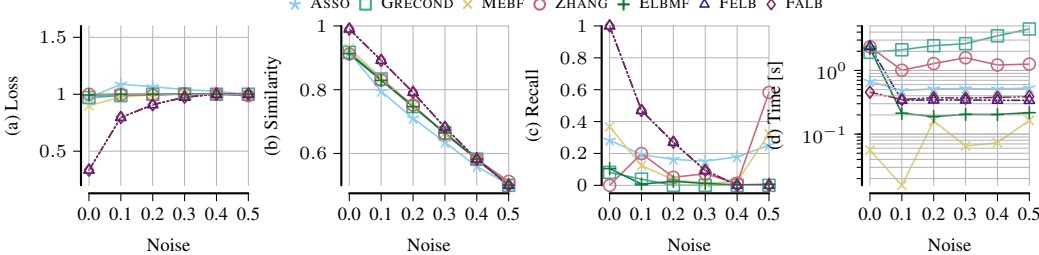

Figure 2: FELB and FALB are robust against noise. We show the loss, recall, similarity, and elapsed runtime ($s/C$) for synthetic data with varying levels of destructive XOR noise.

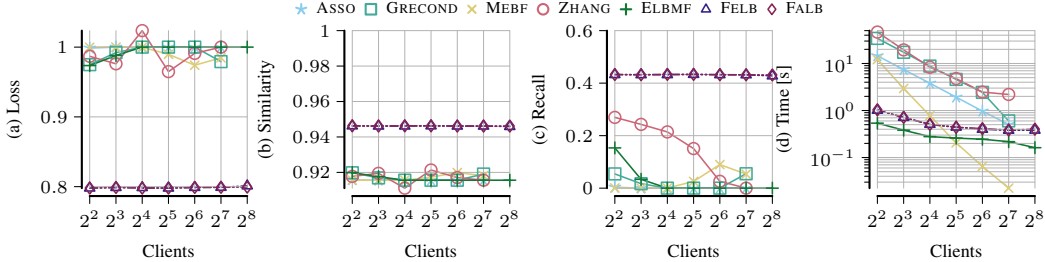

Figure 3: Under *data scarcity*, FELB and FALB perform well across client counts. For a fixed-size dataset and an increasing number of clients, we show loss, similarity, recall, and runtime ($s/C$).

Specifically, we generate synthetic matrices with $2^{16}$ rows and $500$ columns under $5\%$ destructive noise, which we then distribute uniformly at random to the clients, whose number varies from $2^2$ to $2^9$. We depict our results in Fig. 3 , observing that FELB and FALB scale well up to the (per-client) low-observations regime, resulting in almost identical reconstructions, which always significantly outperform ASSO, GRECOND, MEBF, ZHANG, and ELBMF in terms of loss, recall, and similarity. As further elaborated in Appendix D.3, this holds for all three aggregation strategies. In Fig. 3(d), we see that the runtime $[s/C]$ of FELB and FALB is not affected by the reduction in observations per client and almost stays constant, unlike the runtime of ASSO, GRECOND, MEBF, and ZHANG, whose *relative* runtime reduces significantly with the size of the client data, converging towards the runtime of FELB and FALB at the expense of reconstruction quality.

**Data Abundance.**    Second, we study the case in which data is abundant. That is, we assume that *data will increase* with an increasing number of clients. Therefore, we generate increasing amounts of synthetic data, which we then distribute uniformly at random to our clients, such that the amount of per-client data stays roughly the same and the larger total amount of data will pose a significant challenge to the scalability of our competing methods.

Specifically, we generate synthetic data under $5\%$ destructive noise with $500$ rows *per client* and $500$ columns in total. For a number of clients varying from $2^2$ to $2^9$, we depict loss, recall, similarity, and runtime in Fig. 3, providing further details in Appendix D.2. We observe that FELB and FALB, which perform almost identically, significantly outcompete federated ASSO, GRECOND, MEBF, ZHANG, and ELBMF regarding all three aggregation strategies in loss, recall, and similarity. Our approaches almost stay constant with the number of clients, with performance remaining high well into the high-client-count regime. Roughly speaking, the per-client runtime ($[s/C]$ in Fig. 3(d)), of most algorithms is constant, except for GRECOND, whose speed improves by shrinking client data.

### 5.1.3 PERFORMANCE UNDER PRIVACY

To empirically ascertain the effect of differential-privacy guarantees on the loss, we add noise to the transmitted factor matrices according to various noise mechanisms. Specifically, we study the effect on FELB and FALB subjected to clipped ($^\dagger$) or regular Laplacian (Lap) and Gaussian (N) noise mechanisms, as depicted in Fig. 5 and Appendix D.4, for varying $0 \le \epsilon \le 2$ and fixed $\delta = 0.05$.

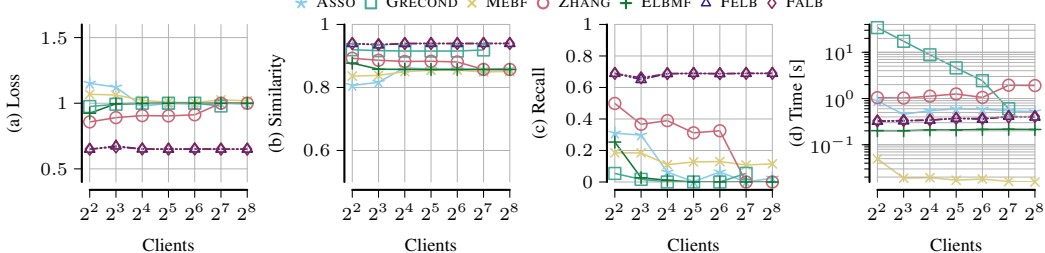

Figure 4: Under *data abundance*, FELB and FALB perform well across client counts. For a growing dataset and an increasing number of clients, we show loss, similarity, recall, and runtime ($s/C$).

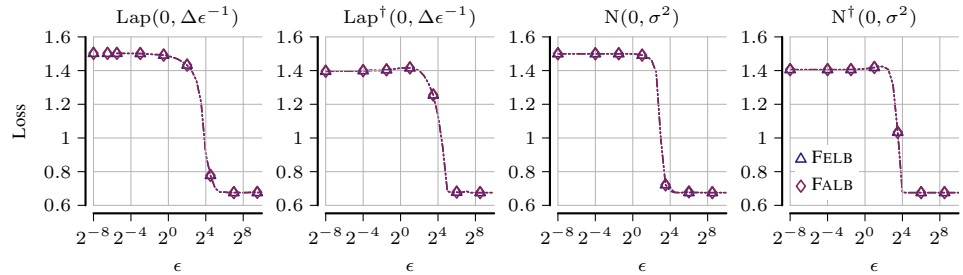

Figure 5: FELB and FALB achieve accurate yet differentially-private reconstructions. For synthetic data, we subject FELB and FALB to four different noise mechanisms: clipped ($\dagger$) or non-clipped Laplacian noise (Lap) or Gaussian noise (N).

The results in Fig. 5, show that both FELB and FALB exhibit nearly identical performance across various noise models. All graphs display a high performance degradation in the low-$\epsilon$ region, followed by a rapid deescalation that subsequently levels off, observing a 'S'-shaped curve, consisting of three phases: Initially, there is obvious performance deterioration in the low-$\epsilon$ domain, followed by a steep, hockey-stick-like descent which eventually stabilizes in the high-$\epsilon$ range. Also noticeable is the increasing 'sharpness' of the hockey-stick-phase under clipping for both mechanisms, showing less smooth reactions to privacy adjustments.

## 5.2 EXPERIMENTS ON REAL-WORLD DATA

Having established the efficiency and precision of our method in factorizing synthetic data, we proceed to assess its effectiveness in handling real-world datasets. For this purpose, we have curated a diverse selection of 10 real-world datasets spanning four distinct domains.

To explore the realm of **recommendation systems**, we have included *Goodreads* (Kotkov et al., 2022) for book recommendations, as well as *Movielens* (Harper and Konstan, 2015) and *Netflix* (Netflix, Inc., 2009) for movie recommendations. To focus on positive ratings, we binarized user ratings, setting ratings $\geq 3.5$ to 1. In the field of **life sciences**, we consider cancer genomics through *TCGA* (Institute, 2005), single-cell proteomics using *HPA* (Bakken et al., 2021; Sjöstedt et al., 2020), and mutation data with *Genomics* (Oleksyk et al., 2015). Specifically, *TCGA* records 1s for gene expressions in the upper 95% quantile, *HPA* designates each observed RNA in each single cell as 1, and *Genomics* indicates the presence of a SNP mutation with a 1. For **social science** inquiries, we investigate poverty *(P)* and income *(I)* analysis using the *Census* (U.S. Census Bureau, 2023) dataset. To binarize, we employ one-hot encoding based on the features recommended by Folktables (Ding et al., 2021). In the domain of **natural language processing**, we focus on higher-order word co-occurrences using ArXiv abstracts from the cs.LG category (Collaboration, 2023). Each paper corresponds to a row whose columns are 1 if the corresponding word in our vocabulary has been used in its abstract. The vocabulary consists of words with a minimum frequency of $1\ \text{‰}$ in ArXiv cs.LG abstracts (*cs.LG R*) and their lemmatized, stop-word-free counterparts (*cs.LG*). We summarize extents, density, and chosen component counts for each real-world dataset in Appendix C, Table 1.

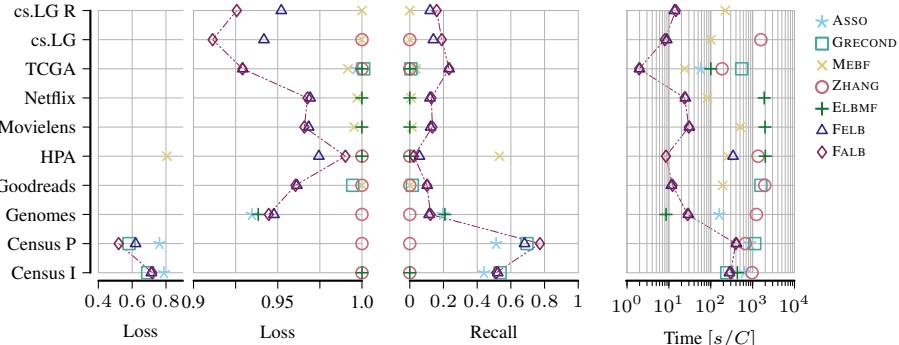

Figure 6: FELB and FALB accurately factorize federated real-world data. We show loss, recall, and runtime ($s/C$) of aggregated ASSO, GRECOND, MEBF, ZHANG, ELBMF, as well as federated FELB and FALB, applied to real-world data. For visual clarity, we split the loss panel into its two separate regimes.

For a fixed client-count of 20, we experimentally compare federated ASSO, GRECOND, MEBF, ELBMF, and ZHANG, as well as FELB and FALB, across all real-world datasets. The results, encompassing loss, recall, and runtime, are presented in Fig. 6. We observe that FELB and FALB consistently significantly outperform aggregated ASSO, GRECOND, MEBF, and ELBMF in terms of loss, recall, and runtime. This performance gap is notable in all datasets with the exception of the *HPA* dataset, where MEBF—a method motivated by datasets of a similar nature—outperforms FELB and FALB. Across the board, FELB and FALB exhibit near-identical performance, with distinctions observable in the *Census P*, *cs.LG*, *cs.LG R*, and *HPA* datasets. Generally striking is the elevated reconstruction loss consistently yielded by aggregated ASSO, GRECOND, MEBF, ELBMF, and ZHANG across most datasets. As depicted in the right panel of Fig. 6, FELB and FALB substantially outpace ASSO, GRECOND, ELBMF, and they even beat the high-performing methods MEBF, ELBMF, and ZHANG in terms of runtime across all real-world datasets.

In this set of experiments, there is no clear winner between FELB and FALB in terms of accuracy of runtime. To clarify this, we additionally experiment with different component counts in Appendix D.5, illustrating results in Appendix Fig. 17, where we see that FALB tends to outcompete FELB for the same set of hyperparameters and larger component counts.

## 6 DISCUSSION AND CONCLUSION

We introduced the federated proximal-gradient-based FELB for BMF tasks, showed its convergence to a binary outcome in theory, and demonstrated its efficacy in experimental practice. We provided a variant called FALB, whose practical performance outcompetes FELB on many real-world datasets. Although FELB and FALB perform consistently well, both are first-of-their-kind federated BMF algorithms. As such, they leave ample room for further research.

First, an important research direction investigates the impact of heterogeneous client data on the performance of FELB and FALB, as heterogeneity often hampers federated-learning strategies. Second, although our proximal-aggregation approach performs well for relaxations, it cannot be applied to Boolean matrices directly. Therefore, while we used state-of-the-art aggregation schemes to federate our baseline BMF algorithms, exploring more advanced Boolean aggregation strategies (e.g., more sophisticated voting mechanisms) constitutes an interesting avenue of future research. Likewise, solving an improved aggregation problem, tailored to the intricacies of matrix factorization, could further improve convergence rates and reduce the need for aggregation. As this likely increases the computational burden, investigating federated optimization for complex aggregations, in the spirit of PROXSKIP (Mishchenko et al., 2022), also appears worthwhile.

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

SUPPLEMENTARY MATERIAL

In this Appendix, we provide supplementary information

- regarding our adaptive proximal operator in Sec. A,
- regarding the federation of baseline BMF methods in Sec. B, and
- regarding reproducibility of our experiments in Sec. C.

Furthermore, we provide additional experimental results regarding

- noise robustness in Sec. D.1,
- scalability under data scarcity in Sec. D.2,
- scalability under data abundance in Sec. D.3,
- differential privacy in Sec. D.4, and
- real-world data in Sec. D.5.

## A  ADAPTIVE PROXIMAL OPERATOR

Replacing the non-adaptive (global) regularization rate $\lambda_t$ in Eq. 4 with an adaptive regularization rate $\lambda_t(x)$ leads to the ALB-regularizer

$$R_{\kappa\lambda}^{\text{ALB}}(X) = \sum_{x \in X} \min \left\{ \kappa\|x\|_1 + \lambda_t(x)\|x\|_2^2, \kappa\|x-1\|_1 + \lambda_t(1-x)\|x-1\|_2^2 \right\} . \tag{9}$$

In other words, the core difference between the ELB and ALB regularizers lies in using an adaptive regularization rate rather than the rate $\lambda_t$. Therefore, following the derivation of the ELB proximal operator (Dalleiger and Vreeken, 2022) analogously yields the adaptive proximal operator

$$\text{prox}_{\kappa\lambda}^{\text{ALB}}(x) = \begin{cases} (1 + \lambda_t(x))^{-1}(x - \kappa \operatorname{sign}(x)) & \text{if } x \le \frac{1}{2} \\ (1 + \lambda_t(1-x))^{-1}(x - \kappa \operatorname{sign}(x-1) + \lambda_t(1-x)) & \text{otherwise} \end{cases}, \tag{10}$$

for the sigmoidal function $\lambda_t(x) = \lambda_t[1 - \exp(-\Gamma \cdot x)]^{-1}$, a constant shift of $\Gamma = 10$, and the usual user-defined monotonically-increasing regularization-rate $\lambda_t$. Using a sigmoidal scaling function allows us to adaptively calibrate the strength of $L_2$ pushing toward 0 (resp. 1). The closer a matrix cell $x$ gets to 0 (resp. 1), the stronger the push, which improves convergence in the last stages. We refer to the version of FELB using the ALB regularizer and adaptive proximal operator as FALB.

## B  COMPETITORS

For a given aggregation function (such as rounded averaging (6), majority voting (7), or logical `or` (8)), we summarize the federation strategy of centralized BMF algorithms in Alg. 2.

---

**Algorithm 2:** Aggregated BMF

**Input:** $C$ clients with local matrices $A^1, \ldots, A^C$, local BMF algorithm $\mathcal{A}$, aggregation function
    aggregate

**Output:** local feature matrices $U^1, \ldots, U^C$, global coefficient matrix $\widehat{V}$

1 **Locally at client $i$ do**
2   $U^i, V^i \leftarrow \mathcal{A}(A^i)$
3 **Centrally at server do**
4   receive $V^1, \ldots, V^C$
5   $\widehat{V} \leftarrow \text{aggregate}(V^1, \ldots, V^C)$
6   transmit $\widehat{V}$ to all clients
7 **Locally at client $i$ do**
8   reciving $\widehat{V}$ from the server
9   assign $V^i \leftarrow \widehat{V}$

---

Table 1: Real-world datasets from 4 diverse domains. We show extents, density, and the selected number of components for 10 real-world datasets.

| Dataset | Rows | Cols | Density | Clients | Components |
|---------|------|------|---------|---------|------------|
| Census I | 1630167 | 998 | 0.010 | 20 | 20 |
| Census P | 3271346 | 836 | 0.024 | 20 | 20 |
| cs.LG | 145981 | 14570 | 0.005 | 20 | 50 |
| cs.LG R | 145981 | 25217 | 0.003 | 20 | 50 |
| Genomes | 2503 | 226623 | 0.104 | 20 | 100 |
| Goodreads | 350332 | 9374 | 0.001 | 20 | 50 |
| HPA | 76533 | 20082 | 0.239 | 20 | 100 |
| Movielens | 162541 | 62423 | 0.002 | 20 | 20 |
| Netflix | 480189 | 17770 | 0.007 | 20 | 20 |
| TCGA | 10459 | 20530 | 0.019 | 20 | 33 |

## C   REPRODUCIBILITY

Supplementing the information provided in Section 5, here, we provide hyperparameter choices for FELB and FALB. We use the iPALM optimization approach for FELB and FALB. Because both algorithms exhibited relatively stable performance fluctuations when it came to tuning, we used the same set of hyperparameters for each experiment and each dataset, thus omitting the commonly necessary hyperparameter tuning step. In all experiments, we used the regularizer coefficients $\lambda = 0.01$ and $\kappa = 0.01$, a regularization rate $\lambda_t = \lambda \cdot 1.005^t$, an iPALM inertial parameter $\beta = 0.01$, a maximum number of iterations of 100, and a number of local rounds per iteration of 10. For ELBMF, we re-use most of the parameters above. However, we provide ZHANG and ELBMF with a larger iteration limit of 1 000, multiplying FELB's local rounds by its iteration count.

### C.1   OBTAINING BOOLEAN MATRICES FROM ZHANG'S FACTORIZATION

The relaxation-based binary matrix factorization of ZHANG (Zhang et al., 2007) does not necessarily yield Boolean factors upon convergence. Furthermore, this method yields matrices that do not lend themselves to rounding, such that in practice, rounding does not yield desirable results *unless* the rounding threshold is carefully chosen. To choose well-factorizing rounding thresholds, we take inspiration from PRIMP (Hess et al., 2017), searching those thresholds that minimize the reconstruction loss,

$$\sum_{c \in [C]} \|A^c - [U^c_{ij} \geq \alpha]_{ij} \circ [V^c_{ij} \geq \beta]_{ij}\| ,$$

from the equi-distant grid between $1 \times 10^{-12}$ and 1 containing 100 points in each direction.

## D   ADDITIONAL EXPERIMENTS

Complementing the discussion in Sec. 5, here, we show additional results for ASSO, GRECOND, MEBF, ELBMF, and ZHANG, as well as FELB and FALB, for all experiments. We focus on the quantification not present in the main body of this paper. That is, we mostly address the *logical* or and *rounded average* aggregations for baseline BMF methods, and we further report the partially postponed quantification in terms of *precision* and *recall*.

### D.1   ROBUSTNESS REGARDING NOISE

Supplementing the experiments in Sec. 5.1.1, here, we report our results for *logical* or (8) and rounded average (6) aggregations. In Fig. 7, we compare FELB and FALB to their competitors under *logical* or aggregation. The trajectories of all methods look similar to those shown in Fig. 2, however, the losses of ASSO, GRECOND, MEBF, ZHANG, and ELBMF increase relative to voting. Compared to *logical* or, the performance of *rounded averaging*, as depicted in Fig. 8, is generally better, especially when considering recall or the 'tighter' similarity trajectory. Noticeable, however, is the loss, which is much closer to 1 for ASSO, GRECOND, MEBF, ZHANG, and ELBMF, regardless of the noise-level.

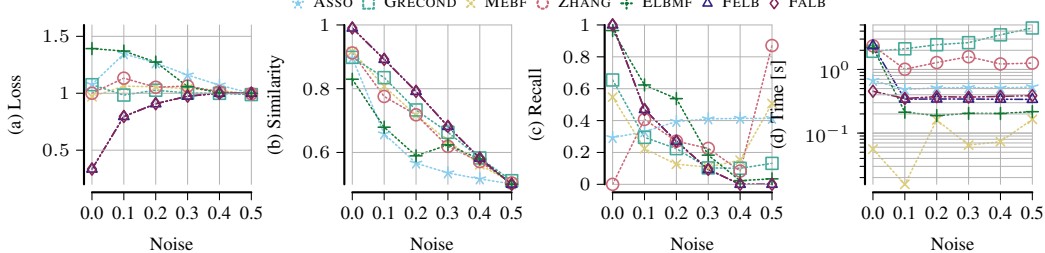

Figure 7: FELB and FALB are robust against noise, outperforming *logical* `or` BMF algorithms. We show the loss, recall, similarity, and elapsed runtime ($s/C$) for synthetic data with varying levels of destructive XOR noise, compared to *logical* `or`-aggregated BMF algorithms.

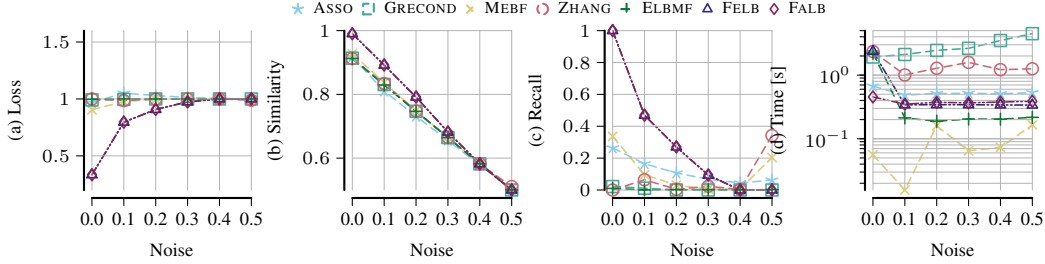

Figure 8: FELB and FALB are robust against noise, outperforming *rounded average* BMF algorithms. We show the loss, recall, similarity, and elapsed runtime ($s/C$) for synthetic data with varying levels of destructive XOR noise, compared to *rounded average*-aggregated BMF methods.

## D.2 SCALABILITY REGARDING CLIENTS UNDER DATA SCARCITY

In this section, we provide additional results for the *data scarcity* experiments laid out in Sec. 5.1.2, reporting results for *logical* `or` (8) and *rounded average* (6) aggregations. In Fig. 9, we observe that the factorization accuracy in terms of *logical* `or` drops in comparison to Fig. 10 and Fig. 3. We see that regardless of the aggregation method, FELB and FALB outcompete the baseline methods significantly in terms of loss, recall, and similarity.

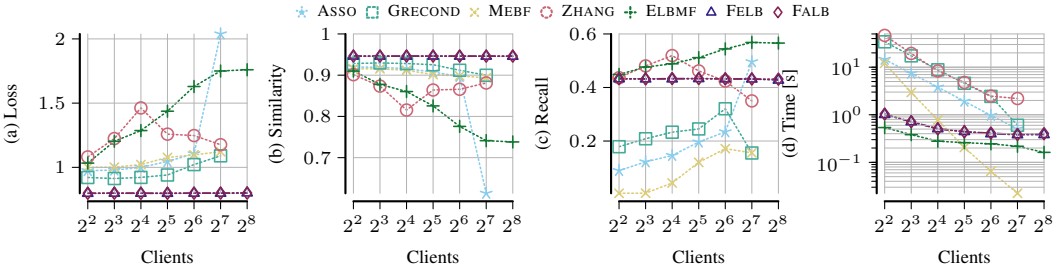

Figure 9: Under *data scarcity*, FELB and FALB perform well across client counts, outperforming *logical* `or`-aggregated BMF methods.

## D.3 SCALABILITY REGARDING CLIENTS UNDER DATA ABUNDANCE

Analogous to Sec. D.2, we also show results for the *data abundance* experiments from Sec. 5.1.2, depicting our results for *logical* `or` (8) and *rounded average* (6) aggregations. In Fig. 11 and in Fig. 12, we see that FELB and FALB are considerable improvements over ASSO, GRECOND, MEBF, ZHANG, and ELBMF, independent of the choice of aggregation. We further observe that our methods

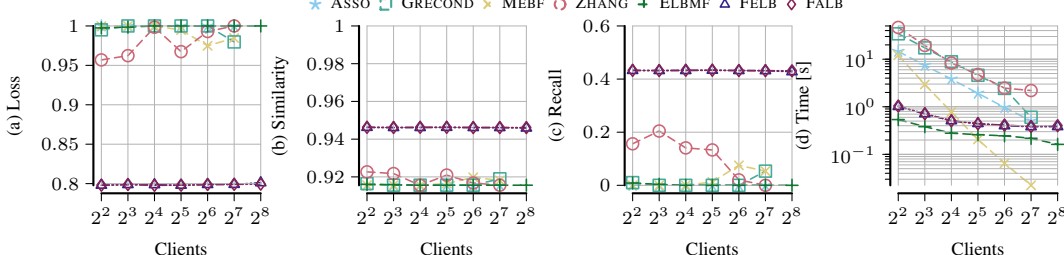

Figure 10: Under *data scarcity*, FELB and FALB perform well across client counts, outperforming *rounded average*-aggregated BMF methods.

almost exhibit a constant trajectory, which means that their performance and accuracy scale well with the number of clients in the abundant-data regime.

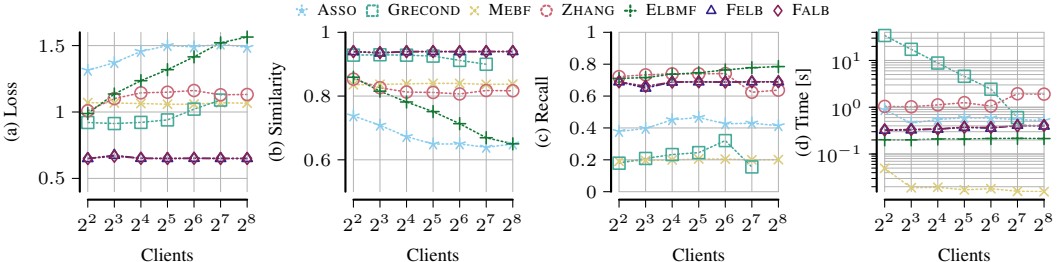

Figure 11: Under *data abundance*, FELB and FALB perform well across client counts, outperforming *logical* `or`-aggregated BMF methods.

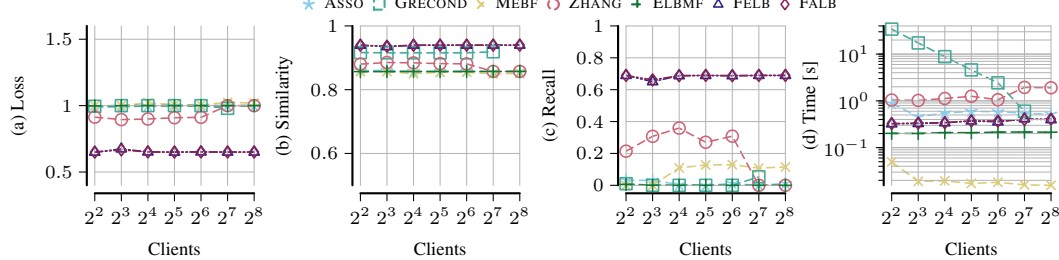

Figure 12: Under *data abundance*, FELB and FALB perform well across client counts, outcompeting *rounded average*-aggregated BMF methods.

## D.4 DIFFERENTIAL PRIVACY

Complementary to the experiments in Sec. 4.2—where we estimated the sensitivity by the maximum number of 1s in the input data— we now want to study the impact of having a pessimistic sensitivity estimate. To this end, we parameterize the noise mechanisms with the upper-bounded *sensitivity* as the column-count of the input matrix, depicting the implications in Fig. 13. There, we see that the loss drops to a low level that is similar to results from the main paper, however, it does so for significantly higher values of $\epsilon$. We see this trend regardless of the noise mechanism, showcasing the impact on higher-sensitivity estimates.

**Boolean Noise Mechanisms.** Both the Laplacian and Gaussian noise mechanisms operate on $\mathbb{R}$-valued matrices. However, clients built upon ASSO, GRECOND, MEBF, ELBMF, and ZHANG transmit binary matrices to the server. Consequently, we are precluded from employing these conventional

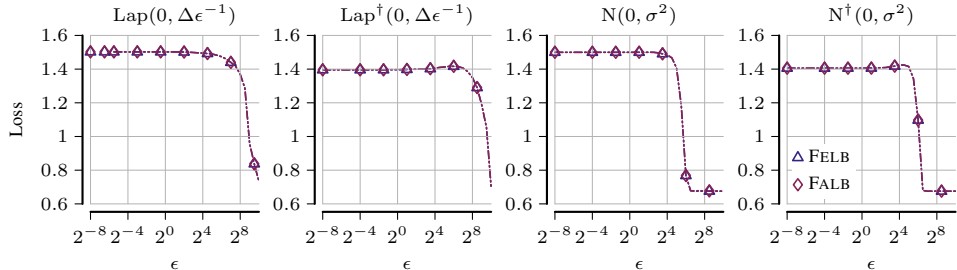

Figure 13: FELB and FALB achieve accurate yet differentially-private reconstructions. For synthetic data, we subject FELB and FALB to four different noise mechanisms: clipped ($^\dagger$) or non-clipped Laplacian noise (Lap) or Gaussian (N) noise, which are parameterized by the upper-bounded *sensitivity* using the number of columns of the input matrix.

noise mechanisms in this context. In order to ensure differential privacy for such clients, we make use of different noise mechanisms that are tailored to binary matrices. Specifically, we consider the XOR mechanism (Ji et al., 2021) parameterized by the input-column count as an upper bound on the sensitivity. Illustrating the differential-privacy impact on the loss in Fig. 14, we note that the losses barely react to varying values of $\epsilon$ and almost immediately plateau, regardless of the federation strategy or underlying BMF algorithm.

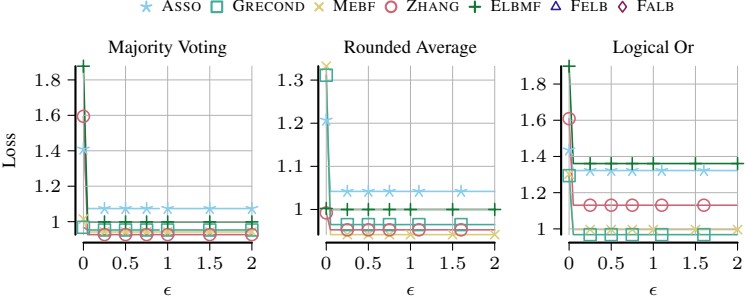

Figure 14: Competing federated BMF methods struggle with noise mechanisms. Federated ASSO, GRECOND, MEBF, ELBMF, and ZHANG do not achieve usable and private factorizations under *XOR* or *Boolean* noise mechanisms, which are parameterized by the input-column count as an upper bound on the sensitivity.

## D.5 REAL-WORLD EXPERIMENTS

Finally, we compare the performance on real-world data, in terms of the *logical* `or` and the *rounded average* aggregation strategies for baseline BMF methods, which are omitted from the main paper.

To analyze if FELB and FALB keep their performance gap relative to differently aggregated BMF algorithms, we compare against *logical* `or` in Fig. 15 and *rounded averaging* in Fig. 16. In both figures, we see that our methods outperform the baselines for each aggregation strategy. We notice from Fig. 15 that *logical* `or` factorizations tend to have the highest loss among all aggregation strategies, which, put informally, is caused by their tendency to 'overfit' the 1s in the data.

To further ascertain the performance differences between aggregated baseline methods on the one hand and FELB and FALB on the other hand, we compare their losses for varying component counts. More precisely, for a fixed client count of 20, we compare federated ASSO, GRECOND, MEBF, ELBMF, and ZHANG, as well as FELB and FALB, parameterized by different configurations, reporting losses for 100 (left), 200 (center), and 500 (right) components in Fig. 17. There, we observe that FELB and FALB improve considerably with the increasing number of components, enabling factorizations with lower losses. Especially interesting is the increasing performance gap between FELB and FALB for larger component counts. That is, for higher component counts, FALB is almost always the

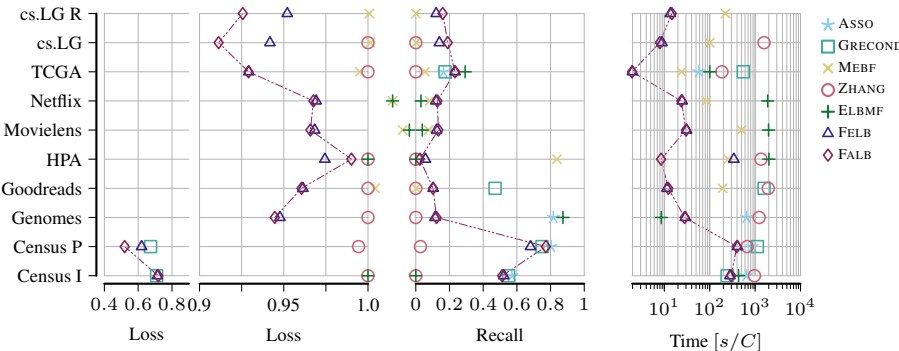

Figure 15: FELB and FALB accurately factorize federated real-world data. We show loss, recall, and runtime ($s/C$) of *logical* `or`-aggregated ASSO, GRECOND, MEBF, ZHANG, ELBMF, as well as federated FELB and FALB, applied to real-world data.

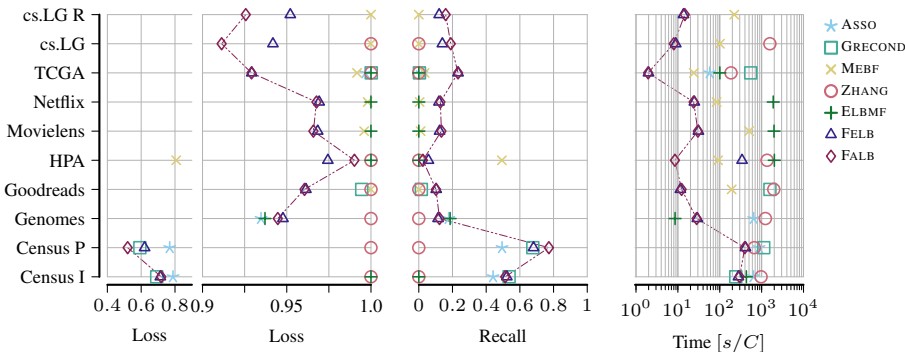

Figure 16: FELB and FALB accurately factorize federated real-world data. We show loss, recall, and runtime ($s/C$) of *rounded average*-aggregated ASSO, GRECOND, MEBF, ZHANG, ELBMF, as well as federated FELB and FALB, applied to real-world data.

best-performing algorithm, significantly outperforming FELB, which takes the second place with large margins to both FALB and the aggregated baselines. We further notice the considerably smaller impact of component counts on losses for aggregated baseline BMF algorithms.

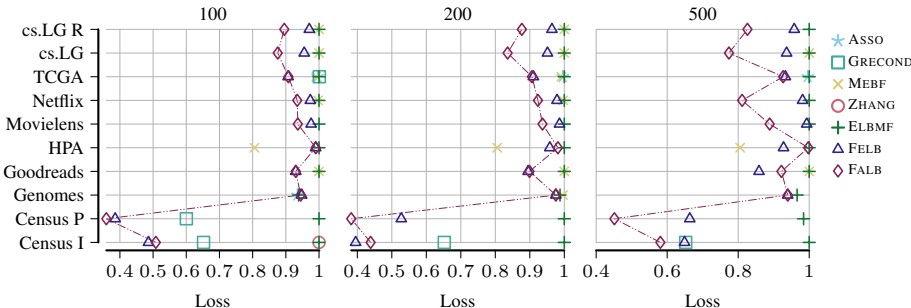

Figure 17: FELB and FALB efficiently and accurately factorize real-world data for increasing numbers of components. FALB outperforms FELB by a large margin for high component counts. We show the loss of federated ASSO, GRECOND, MEBF, ZHANG, and ELBMF, as well as FELB, and FALB, applied to real-world datasets, for 100, 200, and 500 components.

