# OpenReview forum: "Federated Binary Matrix Factorization using Proximal Optimization"
_ICLR.cc/2024/Conference — Submitted to ICLR 2024_

### Official Review · Reviewer_d72F · 2023-10-17

**Soundness:** 3 good
**Presentation:** 2 fair
**Contribution:** 2 fair
**Rating:** 5
**Confidence:** 3

**Summary:**

This paper studies the Boolean matrix factorization problem, where you are given a Boolean matrix $A$, and you need to find two low-rank matrices $U$ and $V$ such that $A\approx U\circ V$, where the operations are over the Boolean semiring. This problem has many practical applications. However, it is NP-hard to solve it exactly. This paper considers a heuristic approach based on the proximal gradient descent. The main contribution of this paper is to propose a federated proximal-gradient-descent algorithm for BMF (FELB) and its adaptive version (FALB). In theory, it proves the convergence of these algorithms and shows some differential privacy guarantees. In experiments, it tests the new algorithms in several different datasets and demonstrates advantages in accuracy and efficiency over prior approaches.

**Strengths:**

The algorithm is the first of its kind in the field of federated BMF. It establishes a new benchmark for future studies in this area. Additionally, the algorithm works well empirically, which could be useful in many real-world applications. The experimental results are comprehensive and clearly stated in this paper.

**Weaknesses:**

The technical novelty of this paper is limited. The idea of using proximal gradient descent has been previously used by Dalleiger and Vreeken (2022). This paper merely transforms the previous algorithm into a federated version. The non-trivial part is the aggregation procedure. However, it is a heuristic and also increases the time complexity for the server. The paper does not theoretically show the convergence rate of the algorithm. Furthermore, the analysis relies on a strong assumption of the loss function. The writing of this paper could be improved. For example, more intuition for the proximal operator should be provided. The current manuscript is not self-contained, and it could be difficult for people outside of this field to understand.

**Questions:**

* Eq. (2): why $A_{ij}$ or $[U\circ V]_{ij}$? It seems to be xor.

* After Eq. (3): the definition of $R$ is confusing. It maps $\mathbb{R}^{n’\times m’}$ to $\mathbb{R}$. However, Eq. (3) applies it to $U$ and $V$, which are matrices with different dimensions.

* Eq. (4): it is unclear what it means for $x\in X$. Is $X$ a set of points, and $x$ is a vector? Or is $x$ a number? And what is $x-1$?

* Page 4: “Without the knowledge of $U_i$, the server cannot estimate specific attributes of individual users (assuming sufficiently large client datasets).” Can the sever learn $U$ from $A$ and $\hat{V}$? Is it proved in this paper to rule out such an attack? If so, a reference should be added here.

* Proposition 1: the symbol R should be replaced by some other symbol, as it has been used to denote the regularizer.

* Page 5, line 4: is it a typo for $\nabla_U\nabla_U$?

* Proposition 1: even if the gradient tends to zero, the loss function may not tend to zero without further assumption on the loss function. Is the loss function considered here convex? And how do we justify the assumption on the bounded gradient?

* Section 4.2: the differential privacy claims should have a formal statement with proof.

* Page 7: the index of the summation should be stated explicitly.

* (Chen et al. 2022) considers the symmetric and sparse version of BMF and shows a combinatorial algorithm for the average-case instances. It is interesting to see whether the method in this paper can be generalized to this regime.

Chen, Sitan, et al. "Symmetric Sparse Boolean Matrix Factorization and Applications." 13th Innovations in Theoretical Computer Science Conference (ITCS 2022).

---

> ### Author Response · Authors · 2023-11-21
> **Reply to Reviewer d72F (part 1)**
>
> Dear reviewer d72F,
>
> Thank you for the kind words, for pointing-out the strength of our work, and for the constructive feedback and extensive list of interesting questions. We address the remaining open points in the following.
>
> **W1) Significant Contribution**
>
> While we draw upon the work of Dalleiger et al. in Boolean matrix factorization, our enhancements are substantial. Firstly, we introduce a novel regularization scheme that surpasses the effectiveness of Dalleiger et al.'s methodology. Secondly, our methodologies represent the inaugural and sole truly federated Boolean matrix factorization algorithm, bridging a crucial research gap and meeting practical demands. Thirdly, we furnish comprehensive guarantees on convergence.
>
> **W2) Aggregation complexity**
>
> We acknowledge that our proximal aggregation increases the work required from our server. However, the proximal operator is not inherently costly to compute. In fact it can be implemented as a single element-wise matrix operation as a GPU kernel. Computational complexity speaking, it is therefore cheaper than matrix multiplication, and will not raise the computational complexity class neither of our server nor of our clients .
>
> Additionally, aggregation does not occur after every update. Given that the proximal operator is applied to the aggregate, the associated increase in complexity remains independent of the number of clients. Consequently, our approach demonstrates scalability with the number of clients, as illustrated in Figures 3 and 4 of the manuscript.
>
> **W3) convergence rate**
>
> The convergence rate trivially follows from our proof of Proposition 1, which will yield a rate of $\mathcal{O}\left(\frac1{\lambda_T}\right)$. That is, setting $\lambda_t = t$ yields a linear convergence rate of $\mathcal{O}\left(T^{-1}\right)$. Note that, by choosing a large lambda (such as $t^2$) we can yield faster convergence, but in practice might lead to a reduction in the relative reconstruction loss.
>
> We thank the reviewer for the interest in this aspect of our algorithm. We will clarify the implications in the final version of our manuscript.
>
> **W4) assumptions regarding loss**
>
> We make use of the standard squared error as loss function and furthermore only make the standard assumption that gradients are bounded. Could you please point out which assumption you deem strong?
>
> **W5) writing, proximal operator intuition**
>
> To improve our presentation and raise the accessibility of our work, we will include more intuitive descriptions in the main body of our paper, regarding—for example—the interplay of gradient descent, proximal operator, regularization, and aggregation.
>
> **W6) self-contained**
>
> To make our paper self-contained, we will expand the details and a complete derivation of our proximal algorithms in the appendix.
>
> **Q1) Eq. (2)**
>
> We deemed this to be more readable, but will revisit the notation.
>
> **Q2) After Eq. (3)**
>
> $m$ and $n$ are meant to be the parameter-dependent row- or column-counts of the input matrix of $R$. To address confusion, we will adapt our notation.
>
> **Q3) Eq. (4): $x$**
>
> $x$ is a single real-valued matrix cell from one of the factor matrices. For example $x = U_{ij}$.
>
> **Q4) Can the server learn I from A and V=?**
>
> Our point is that the server cannot trivially estimate specific attributes about users (i.e., learn $U$ from $A$ and $V$, since factorizations are not unique (nonnegative factorization are not unique, even in case of infinitesimal rigidity [1], and Boolean factorizations can be shown to be not unique by a simple application of the pigeonhole principle, see [2] for further details). We apologize for the imprecision. As stated in Sec. 4.2, we are well aware that a curious server could in principle obtain information about local data from V - one could imagine approaches similar to gradient leakage or membership inference attacks. This is the whole motivation for the differential privacy mechanism in Sec. 4.2 and our empirical evaluation of it in Sec. 5.1.3.
>
>
> **Q5) Proposition 1: the symbol R should be replaced by some other symbol, as it has been used to denote the regularizer.**
>
> Indeed, this is a mistake. We will replace it with another symbol. Thank you!
>
> **Q6) typo**
>
> Good catch!
>
> **Q7) Loss function and gradient**
>
> It is correct that the loss function does not necessarily converge to $0$. Indeed, it is rather unlikely that it does, since it would imply a perfect factorization. Proposition 1 does not assume this. The assumption of a bounded gradient is ubiquitous in the optimization literature and holds, e.g., for Lipschitz continuous loss functions and data from a bounded domain.

---

> > ### Author Response · Authors · 2023-11-21
> > **Reply to Reviewer d72F (part 2)**
> >
> > **Q8) Section 4.2: the differential privacy claims should have a formal statement with proof.**
> >
> > Thank you for this observation. Having a formal statement would indeed highlight the differential privacy claims. Due to space limitations and since the results can be easily derived from (BalleandWang,2018), respectively (Nobleetal.,2022), we decided, however, to keep it simple. We do appreciate your point, though, and will therefore include formal statements and their derivation in the appendix.
> >
> > **Q9) Page 7: sum**
> >
> > Thank you, we will adjust this equation in the final submission.
> >
> > **Q10) Symmetric BMF**
> >
> > This is a great question and the suggested work is interesting. Our work is based on clients with access to a vertical part of the input matrix. That is, rows and columns are treated differently. If the matrix is symmetric, it will ‘break’ this setting. In fact, we made sure that our experimental data will not by accident include symmetric data. Nevertheless, symmetric settings are relevant in general, and hence we will certainly add a discussion of Chen et al.to related work.
> >
> > Again, we thank the reviewer for the valuable feedback. We would like to kindly ask for increasing the score, if we answered the remaining points in a satisfactory way.
> >
> >
> > [1] Krone, Robert, and Kaie Kubjas. "Uniqueness of nonnegative matrix factorizations by rigidity theory." SIAM Journal on Matrix Analysis and Applications 42.1 (2021): 134-164.
> >
> > [2] Breen, Michael, and David Hume. "Properties of a standard form for a Boolean matrix." Linear algebra and its applications 254.1-3 (1997): 49-65.

---

### Official Review · Reviewer_iqWp · 2023-10-28

**Soundness:** 2 fair
**Presentation:** 3 good
**Contribution:** 1 poor
**Rating:** 3
**Confidence:** 4

**Summary:**

The paper proposes a method for federated binary matrix factorization.
It also considers differentially private variants of the proposed method, by adding noise to the local updates (offering local DP guarantees).

**Strengths:**

Presentation and clarity: The paper is overall well-motivated, the ideas are simple and easy to follow, and adequate background is given about binary matrix factorization, federated learning and differential privacy.
The method seems like a reasonable extension of existing methods to the federated setting.

**Weaknesses:**

The contribution of the paper seems limited. This is a straightforward extension of the method of Dalleiger et al. to the federated setting. Each client applies the same method, then the $V_i$ updates are aggregated on the server. The main algorithmic novelty seems to be to how aggregation is done, and the idea was to apply the proximal operator (the *same one used by Dalleiger et al.*) to the mean of the updates.

The theoretical result (convergence of the iterates in Proposition 4.1) has several issues.
First, the proof is wrong. It is argued that the gradients tend to 0, thus the iterates converge. This is not true (think of the sequence $x_k = 1/k$).
Even if the proof were to be corrected, the result is still very weak. It only claims that the process converges (to a binary solution) without saying anything about the limit's utility (for example whether this limit is close to the solution). Such a result can be obtained by a trivial algorithm (for example one that uses a step size of 0 after a finite number of steps).

The paper also seems completely unaware of the extensive literature on private matrix completion (it is claimed in the related work that "algorithms for matrix factorization seek computational efficiency while disregarding privacy aspects"). The problem has been studied (without the binary constraints) in a number of papers both in the central and federated settings [1-4]. Even though the exact setting is different (due to the binary constraints) the methods are closely related, and several ideas used in this paper were already proposed in prior work. For example, this paper proposes "partial sharing" (sharing the V factor but keeping the U_i factors local to each client), but this exact scheme was used in [1-5] (in the privacy literature this is referred to as joint-DP or sometimes as the billboard model of DP).

Other comments:
- What is the motivation to add binary constraints in recommender systems? This does not seem to be a common practice. (Related to this point: some of the recommendation data sets used in experiments were used in prior work on private matrix completion, without binary constraints. It would be good to compare.)
- In section 4, it is claimed that "Without the knowledge of $U_i$, the server cannot estimate specific attributes of individual users". This is not true. For example access to gradients can leak information about the client's data. This has been shown via reconstruction attacks (for example in [6] which looked specifically at matrix completion).
- Privacy-related experiments: it seems that in all cases, for epsilon up to roughly 10, the loss exceeds 1 (which if I understand the loss definition correctly, means the learned solution is worse than even a trivial solution of all 0). This seems surprising, it basically means that learning is impossible except for very large epsilon. This might indicate that the method has not been properly tuned. Perhaps the authors should carefully revisit these experiments.
- It is mentioned in the appendix that sensitivity was estimated by the maximum number of 1s in the input data. But note that this is a private quantity, and it should be estimated privately (not computed exactly), or you can perhaps set a maximum number then subsample each client to this number.
- [a comment that does not affect my evaluation] the authors included their code in the supplementary material, and this is commendable. However, including code without comments or instructions on running experiments makes it virtually impossible to reproduce the results. Ideally, one should include at least a readme file with instructions on how to run the methods (and how to get the data).

1. Chien et al. Private alternating least squares: Practical private matrix completion with tighter rates. ICML 2021
2. Jain et al. Differentially private model personalization. NeurIPS 2021
3. Wang et al. Differentially Private Matrix Completion through Low-rank Matrix Factorization. AISTATS 2023
4. Shen et al. Share your representation only: Guaranteed improvement of the privacy-utility tradeoff in federated learning. ICLR 2023
5. Singhal et al. Federated reconstruction: Partially local federated learning. NeurIPS 2021
6. Chai et al. Secure Federated Matrix Factorization. IEEE Intelligent Systems, 2021

========= Post rebuttal =========
I thank the authors for their response regarding related work, sensitivity, and reconstruction attacks.

Regarding the convergence proof: unfortunately the response indicates a fundamental misunderstanding of convergence. It is claimed in the paper that "FELB converges to a stable binary solution". Convergence was never proved. What was shown instead is that $\|\eta_t\nabla F\|$ converges to 0. Notice that what converges to 0 is the gradient *scaled by the learning rate* (indeed proving this is trivial, as the gradient is assumed to be bounded, and the learning rate is assumed to tend to 0).
This does not prove anything about convergence of the gradient, nor the trajectory.

Regarding the relative loss: I understand that it can exceed 1. My point is that a relative loss exceeding 1 indicates a solution that's essentially useless, as even the trivial 0 solution would achieve a loss of 1. This indicates that the private method is not learning much except for very large epsilon. I encourage the authors again to revisit these experiments more carefully and investigate this poor performance.

I maintain my initial assessment.

**Questions:**

- I invite the authors to reflect/comment on connections to prior work mentioned above.
- Do the authors think they can strength the theoretical analysis?

---

> ### Author Response · Authors · 2023-11-21
> **Reply to Reviewer iqWp**
>
> Dear reviewer iqWp,
>
> Thank you for recognizing and highlighting the quality and clarity of our presentation, as well as the relevance of our work. Next, we address your concerns in detail.
>
> **W1) straightforward extension**
>
> Although we make use of work from Dalleiger et al. on Boolean matrix factorization, our improvements are significant. First, we contribute a novel regularization scheme that clearly outperforms Dalleiger et al.’s approach. Second, our approaches are the first and only truly federated Boolean matrix factorization algorithm in existence, filling an important research gap and addressing real-world needs. Third, we provide rigorous guarantees in terms of convergence.
>
> **W2) Proof**
>
> We respectfully disagree with the comment that the proof is incorrect. The proof is, in fact, correct. There seems to be a misunderstanding in what approaches zero. It is not the gradient itself, as the reviewer seems to have understood, but rather, we assume that the gradient is limited. It is the gradient component, as opposed to the regularizer, that diminishes towards zero. The proof demonstrates that the gradient is bounded by $R$, and since $\eta_t\propto \lambda_t^-1\rightarrow 0$ for $t\rightarrow \infty$, the limited gradient multiplied by a decreasing learning rate tends to zero. We apologize if the wording caused any confusion, and will rewrite this part to avoid this for any future reader.
>
> We also respectfully disagree that a convergence result is without value if the limit’s utility is not demonstrated. In non-convex optimization, such as for deep learning or federated deep learning, it is common practice to illustrate that $\mathbb{E}[\nabla_w F(w)]$ approaches zero, signifying the attainment of a local minimum or saddle point. These convergence results are highly valuable, although they do not imply the usefulness of that limit.
>
> **W3) Matrix Completion**
>
> We thank the reviewer for making us aware of related work.
> We acknowledge that there is related prior art on partially sharing matrices (joint-DP) for federated matrix completion and will reflect this in our related work section.
> However, federated matrix completion methods are only complementary to our problem.
> As you also already point out yourself, Algorithms [1-5] focus on non-Boolean data.
> Imposing Boolean constraints on the resulting factorization is central to our problem setting.
>
> **Q1) Datasets**
>
> Boolean data is ubiquitous in real-world settings, such as gene mutation data.
> In many recommender applications, the goal is to distinguish between good or bad recommendations. Factorizing a binarized data matrix with two Boolean factor matrices,  will result in sparse (Miettinen et al. 2006), informative, and easily interpretable factorizations by default.
>
> Our datasets are derived from commonly-used and well-known benchmark datasets.
> However, as we require Boolean inputs, we binarized our datasets first.
> Comparing aforementioned matrix completion algorithms with our method, would not only compare the performance of different tasks, but would also introduce a bias in favor of our algorithm, as—unlike our approaches—the completion algorithms are not specialized towards Boolean data.
>
> **Q2)**
>
> The claim is that the server cannot trivially estimate specific attributes about users, since factorizations are not unique (nonnegative factorization are not unique, even in case of infinitesimal rigidity [1], and Boolean factorizations can be shown to be not unique by a simple application of the pigeonhole principle, see [2] for further details). We apologize for the imprecision. As stated in Sec. 4.2, we are well aware that a curious server could in principle obtain information about local data from V - one could imagine approaches similar to gradient leakage or membership inference attacks. This is the whole motivation for the differential privacy mechanism in Sec. 4.2 and our empirical evaluation of it in Sec. 5.1.3.
>
> **Q3) Relative Loss**
>
> The relative loss can exceed $1$, if the reconstruction consists of more $1$s than the input.
>
> **Q4) Sensitivity**
>
> Thank you for this great observation. Indeed, we have used a very straight-forward estimation of sensitivity in our experiments, that reveals the maximum number of $1$s in the dataset. Using a differential privacy mechanism to share local maxima is a great idea. Setting a global maximum is a neat practical trick, but requires some domain knowledge to set it right - too low and we have to distort data substantially by subsampling, too large and we decrease the quality due to excessive noise.
>
> **Q5) Code**
>
> Of course, we will submit a complete, fully documented, and easily usable reproducibility package, that comes as an archive with DOI and a publicly available and maintained GitHub repository.
>
> We thank the reviewer for the constructive feedback that clearly aims to improve our work. In case we answered your questions in a satisfactory way, please do consider raising your score accordingly.

---

### Official Review · Reviewer_nPms · 2023-11-01

**Soundness:** 3 good
**Presentation:** 3 good
**Contribution:** 3 good
**Rating:** 5
**Confidence:** 3

**Summary:**

The authors present a novel method called Felb, which is a federated binary factorization algorithm based on proximal gradients along with the adaptive alternative Falb.

**Strengths:**

- the authors bring a novel algorithm designed for binary matrix factorization. The previous federated factorization algorithms were not for the binary/boolean matrices.
- the paper is well-written, easy to follow.

**Weaknesses:**

- The contribution is limited to the binary matrices, unless a further performance analysis on the non-binary matrices are provided. (the most of the datasets selected are by default not boolean matrices but converted to boolean)

- The potential disadvantages of having a proximal operator is not being discussed.

- The methods compared could be discussed further, for instance in Figure 6. the loss values for most of the models seem to be stuck at 1. What is the reason for that? What are the parameters used for these methods? How are they implemented? It might be possible that this is a bug. Similarly the recall values are zero. This is skeptical.

Please refer to the Questions section for the minor questions.

**Questions:**

- We demonstrate that our approach converges for a strictly monotonically increasing regularization rate.

Could authors elaborate more on this? How is this useful? What is the rate? Is there an ablation study on different rates?

- Most of the recent research shows that preserving privacy inevitably brings a reduction in accuracy. The authors claim that the accuracy of the method is even higher than the methods that do not have a privacy component. What is the trade-off? computational complexity? scalability? time complexity? difficulty in implementation?

- What are the advantages that the proximal gradients bring to the problem?

- Did the authors do any comparison with the new federated matrix factorization (Du et al., 2021) and federated non-negative matrix factorization (Li et al., 2021).

- Figure 6, is there a logical order of the datasets?

- Where is the differential privacy stated in the formulation?

- Is the lambda different for u and v? (Algorithm 1, line 4 and 5)

**Details Of Ethics Concerns:**

This paper does not need ethics review.

---

> ### Author Response · Authors · 2023-11-21
> **Reply to Reviewer nPms (part 1)**
>
> Dear reviewer nPms,
>
> Thank you for your concise summary, for highlighting the quality of our writing, and for recognizing the novelty of our algorithms. We appreciate your valuable feedback and questions that will help us to further improve our work. Next, we address the remaining points.
>
> **W1)  Limited to binary data**
>
> We do not agree that our scope is limited. Our scope is merely different from the scope of non-Boolean matrix factorization. Boolean data is ubiquitous in real-world settings, such as gene mutation data. In many recommender applications, the goal is to distinguish between good or bad recommendations. Factorizing a binarized data matrix with two Boolean factor matrices,  will result in sparse, informative, and easily interpretable factorizations by default.
>
> That being said, it would be unfair to experimentally compare binary matrix factorizations of discretized inputs to non-binary matrix factorizations of non-discretized inputs.
> Applying non-Boolean matrix factorization, e.g. NMF, neither gives such guarantees and often simply does not make sense (e.g. gene activations).
>
> In our experiments, we have selected a mix of easily-recognizable datasets for the MF community and a mix of datasets where the discretization yields results that are sparser and easier to interpret. That is, in those applications it is important to gain insight into which features tend to be jointly present (e.g. yes/no recommendations) resp. for which it primarily matters whether the value is high or low, and the actual values matter much less (e.g. gene expression data).
>
> **W2) advantages of a proximal operator**
>
> Approaching binary matrix factorization in terms of a proximal algorithm has similar benefits as approaching NMF with proximal gradient descent. That is, by separating loss and binary regularization, we can take a much longer “unregularized” gradient step than the steps taken by e.g., multiplicative update rules (MUR). This will usually result in significantly faster convergence rates in comparison to MUR.
>
> The downside is that we need to compute the proximal step. This, however, is done by a single element-wise matrix operation via an efficient GPU kernel in O(n*k). We will clarify the advantages of our proximal approach in the updated version of the manuscript.
>
> **W3) competitors, parameters**
>
> We share the desire for having a rigorous and extensive evaluation. We do believe that our evaluation fits this standard well.
>
> **W3.1) Regarding reproducibility**
> - The parameters can be found and are discussed in the “Reproducibility” section in the appendix.
> - The implementations for the competing BMF algorithms are publicly available from their respective authors.
>
> **W3.2) Regarding results**
>
> - We consider a _relative_ loss, normalized by the Frobenius norm of the input matrix, rather than an absolute loss, as this allows for more insightful comparison across datasets.
> - During the empirical evaluation, we extensively studied the performance, looked into reconstructions, and validated the correctness of all methods, including the “competing” algorithms. On synthetic data, we observe a different trend in the performance of our “competitors” on synthetic data, which is evidence for this. On real-world data, those algorithms do not allow for an accurate reconstruction in terms of shared coefficient (component) matrices. Because the algorithms perform as expected locally, we discovered that widely-used aggregation strategies fail to address the problem well.
> We therefore proposed our two methods to address this problem.
>
> **Q1)  Usefulness of a Regularization Rate**
>
> The regularization rate allows us to converge to a Boolean factorization in the limit.
> This is a crucial aspect of the convergence proof of our algorithm.
> The exact rates are given in the “Reproducibility” section of the Appendix.
>
> Because FELB and FALB exhibited robust and stable performance behavior, under different yet sensible regularization rates, we skipped a full-blown ablation study for brevity.
> Therefore, we chose a single exponentially-growing regularization rate, which is $\lambda_t = \lambda · 1.005^t$. As is the case for any method, adversarially chosen parameters will of course yield bad results.
>
> **Q2) privacy**
>
> The enhanced privacy of occluding observation-specific factors not only reduces the computational complexity on the server-side but also the amount of data that needs to be transmitted. To guarantee differential privacy, we use an additive noise mechanism, which _reduces_ the accuracy—as expected—with regards to having no additive noise.

---

> > ### Author Response · Authors · 2023-11-21
> > **Reply to Reviewer nPms (part 2)**
> >
> > **Q3) Advantages of proximal gradients**
> >
> > By separating the optimization into a (non-regularized) gradient step and a proximal projection, we take significantly larger gradient steps that will result in faster convergence rates (in comparison to multiplicative update rules)
> > Furthermore, this separation allows for computing the gradients (per factor matrix) using a  known closed form solution, which leads to speedups and performance gains in comparison to, e.g. auto differentiation.
> >
> > **Q4) NMF**
> >
> > We haven’t compared against Federated NMF, because (i) we are after factorizing Boolean matrices and (ii) we are interested in a specific privacy model that keeps the first factor matrix private.
> >
> > **Q5) Order of datasets**
> >
> > We are happy to consider a reorganization of our partial dataset ordering in our camera ready version.
> >
> > **Q6) Where is the differential privacy stated in the formulation?**
> >
> > In Section 4.2, we show that using the Laplace mechanism with mean $0$ and variance $\Delta\epsilon^{-1}$ achieves $(\epsilon, 0)$-differential privacy. and using the Gaussian mechanism with mean $0$ and variance $\sigma=\Delta\epsilon^{-1}\sqrt{2\log\frac{5}{4\delta}}$ achieves  $(\epsilon, \delta)$-differential privacy.
> >
> > **Q7) Different lambdas per factor**
> >
> > For brevity, we choose the same lambda to parameterize all matrices. This can easily be adapted if the need arises.
> >
> > Having addressed all remaining questions, we thank nPms for the extensive interest in our work. We would like to kindly ask the reviewer to increase the score, if we have answered all questions in a satisfactory way.

---

> > > ### Comment · Reviewer_nPms · 2024-04-03
> > > **Acknowledge**
> > >
> > > I acknowledge that I have read the author's reply and my decision is the same.

---

### Meta-Review · Area_Chair_9uDo · 2023-12-04

**Metareview:**

One of the reviewers pointed out a fatal error in the statement concerning convergence. The authors did not respond to this. Hence, I believe the submission requires a complete rewrite to be technically correct.

**Justification For Why Not Higher Score:**

One of the reviewers pointed out a fatal error in the statement concerning convergence.

**Justification For Why Not Lower Score:**

The score is already rather low due to the presence of an error in the statement/proof.

---

### Decision · Program_Chairs · 2024-01-16

Reject